# Expression Profile and Prognostic Significance of EPHB3 in Colorectal Cancer

**DOI:** 10.3390/biom10040602

**Published:** 2020-04-13

**Authors:** Bo Gun Jang, Hye Sung Kim, Jeong Mo Bae, Woo Ho Kim, Chang Lim Hyun, Gyeong Hoon Kang

**Affiliations:** 1Department of Pathology, Jeju National University School of Medicine, Jeju 63241, Korea; bgjang9633@gmail.com (B.G.J.); iamimza@gmail.com (H.S.K.); 2Department of Pathology, Seoul National University College of Medicine, Seoul 03080, Korea; jeongmobae@gmail.com (J.M.B.); woohokim@snu.ac.kr (W.H.K.); 3Laboratory of Epigenetics, Cancer Research Institute, Seoul National University College of Medicine, Seoul 03080, Korea

**Keywords:** EPHB3, colorectal cancer, immunohistochemistry, tumor suppressor, prognosis

## Abstract

The protein tyrosine kinase Ephrin type-B receptor 3 (EPHB3) is expressed in cells at the base of intestinal crypts, acting as a cellular guide in the maintenance of intestinal crypt architecture. We aimed to investigate the expression profile of EPHB3 in colorectal precancerous lesions and colorectal cancers (CRCs), and assess its prognostic value. EPHB3 expression was higher in CRCs than in normal mucosa and was associated with the intestinal stem cell markers EPHB2, OLFM4, LRIG1, and a proposed cancer stem cell marker, CD44. Enhanced EPHB3 expression significantly declined during the transformation from adenoma to carcinoma and as the tumor invaded into deeper tissue layers. Namely, a substantial reduction of EPHB3 expression was observed in the budding cancer cells at the invasive tumor fronts, which was more extensive than E-cadherin downregulation. In an azoxymethane/dextran sulfate sodium-induced, colitis-associated, CRC model, EPHB3 expression increased along with tumor development. In a large cohort of CRC patients, EPHB3 positivity was observed in 24% of 610 CRCs and was negatively correlated with tumor differentiation, lympho-vascular invasion, and tumor, node, and metastasis stages. EPHB3 was positively associated with microsatellite instability but was associated with neither CpG island methylation, nor with KRAS and BRAF mutations. Notably, EPHB3 positivity was associated with better clinical outcomes, although it was not an independent prognostic marker. Overexpression of EPHB3 in the colon cancer cell line, DLD1, led to decreased cell growth and migration and reduced mitogen-activated protein kinase signaling. Taken together, our data demonstrate the suppressive role of EPHB3 in CRC progression.

## 1. Introduction

Ephrin type-B receptor 3 (EPHB3) is a member of the largest family of receptor tyrosine kinases, Eph [1]. Eph receptors are subdivided into classes A and B based on sequence homology and binding affinity for their membrane-bound ligands [2]. EPHA receptors bind glycosylphosphatidylinositol-anchored ephrin-A ligands and EPHB receptors bind transmembrane ephrin-B ligands [3]. EPHB/ephrin-B interactions are key regulators in diverse physiological and pathological processes associated with development and disease of different organ systems [4,5]. In particular, a wide variety of cancer cells express EPHB receptors and their cancer-related activities are complex and intriguing; their roles can be either tumor suppressive or oncogenic depending on the type of cancer and the cellular context [1,6].

The roles of EPHB3 in colorectal cancer (CRC) development have been characterized using CRC mouse models. EPHB3 is normally expressed by cells in the stem cell niche at the base of intestinal crypts, where EPHB3 and its homologue EPHB2 generate and maintain the architecture of the villus–crypt axis through their interaction with ephrin-B ligands [7]. The loss of EPHB3 accelerates colorectal tumorigenesis and prompts the formation of invasive adenocarcinoma in Apc ^Min/+^ mice, suggesting that EPHB3 suppression represents a critical step in CRC progression [8]. Moreover, EPHB2 and EPHB3-induced compartmentalization can restrict the spread of CRC cells [9]. EPHB3 is downregulated in advanced stages of human CRC [10], which suggests that EPHB3 functions as a tumor suppressor in this context. However, due to the complex functions and bidirectional signaling of the EPH/ephrin system, this hypothesis needs to be verified with a large cohort of human CRC samples. Indeed, despite all the strong evidence supporting a tumor suppressive role for EPHB3 in mouse studies, Xaun et al. recently reported that EPHB3 is an independent prognostic factor for poor survival in CRC patients [11]. Similarly, there are conflicting data regarding the effect of EPHB3 on the progression of non-small cell lung cancer [12,13]. 

While EPHB2 expression is consistently associated with better survival of CRC patients [14,15,16], the prognostic value of EPHB3 in CRCs has not been well characterized. Additionally, no studies have comprehensively assessed the changes in EPHB3 expression from benign to malignant tumors of the colorectum. In this study, we sought to quantify EPHB3 expression in a variety of precancerous lesions and numerous CRC samples, and to determine its prognostic significance. Furthermore, we analyzed alterations in the expression of EPHB3 during various stages of CRC progression: adenoma-to-carcinoma transformation, tumor budding, and lymph node metastasis. We also evaluated EPHB3 expression in a colitis-associated cancer (CAC) model, and its effects on cancer growth and migration were assessed using CRC cell lines.

## 2. Materials and Methods

### 2.1. Participants 

We collected 610 formalin-fixed, paraffin-embedded (FFPE) CRC tissue samples between 2004 and 2006 from the archives of the Department of Pathology at Seoul National University Hospital (SNUH) (Seoul, Korea). Information including patient gender and age; tumor location, histological type, and level of differentiation; evidence of lympho-vascular invasion; American Joint Committee on Cancer/International Union against Cancer (AJCC/UICC) cancer stage (7th edition); time of death; tumor recurrence; and duration of follow-up were obtained by reviewing the clinical and pathological reports. With regard to tumor location, proximal colon was defined as proximal to the splenic flexure (cecum, ascending and transverse colon) and distal colon was defined as distal to the splenic flexure (splenic flexure, descending, sigmoid colon, and rectum). Tumor budding was defined as a single tumor cell or a cluster of <5 tumor cells at the invasive margins. Fifty-nine CRC samples were obtained from the Department of Pathology at Jeju National University Hospital (JNUH) (Jeju, Korea). Twenty-two of the CRC tumors arose from pre-existing adenomas and 37 were ulcerofungating CRCs with lymph node metastases. The histopathologic features of the CRCs were determined by three gastrointestinal pathologists (J.M.B. and G.H.K. assessed the SNUH samples and B.G.J. assessed the JUNH samples). Additionally, 32 paired, fresh colorectal cancer tissues and matched normal tissues were provided by the Jeju National University Hospital Biobank, a member of the National Biobank of Korea, for which informed consent was obtained from all participants. All procedures were conducted in accordance with the 1975 Helsinki Declaration, revised in 2013. The study was approved by the Institutional Review Board of SNUH (C-1502-029-647) and (JNUH 2017-06-029), respectively. 

### 2.2. DNA Isolation

FFPE CRC tissue samples (*n* = 610) were retrieved from the Pathology archives of the SNUH. Genomic DNA isolation was performed as follows: cancer areas (cancer cells > 70% of selected area) were microdissected with surgical blades from 10 μm-thick, unstained tissues. The tissues were digested in lysis buffer (100 mM Tris-HCl (pH 8.0), 10 mM EDTA (pH 8.0), 0.05 mg/mL tRNA and 1 mg/mL proteinase K) at 55 °C for 48 h, followed by a 10-min incubation at 95 °C to inactivate the proteinase K. The DNA was stored at −20 °C.

### 2.3. Microsatellite Instability (MSI) Analysis

The genomic DNA of CRCs was subjected to MSI analysis using the fluorescent multiplex PCR method with five NCI-recommended microsatellite markers (BAT25, BAT-26, D5S346, D17S250, and D2S123). The MSI status of each CRC sample was classified as either MSI-high (≥2 unstable markers of 5), MSI-low (1 unstable marker of 5), or microsatellite stable (no unstable markers).

### 2.4. DNA Methylation Analysis

DNA analysis for the determination of CpG island methylator phenotype (CIMP) status was carried out as previously described [17]. Sodium bisulphite modification of genomic DNA samples was performed for all 1133 CRC specimens. The quantitative measurement of the promoter CpG island methylation of eight CIMP marker genes (CRABP1, CACNA1G, CDKN2A (p16), IGF2, MLH1, NEUROG1, RUNX3, and SOCS1) was performed using MethyLight assay, methylation specific real-time PCR. A CIMP-high tumor was defined when having five or more hypermethylated markers, a CIMP-low tumor was defined when having one to four hypermethylated markers, and a CIMP-negative tumor was defined as having no hypermethylated markers. A hypermethylated CpG island locus was defined when the percentage of the methylated reference (PMR) value was >4. The MethyLight assay for each CIMP marker gene was repeated independently three times, and promoter hypermethylation was defined as a PMR value >4 observed in at least two of three experiments.

### 2.5. KRAS/BRAF Mutation Analysis

KRAS/BRAF mutation analysis was performed as previously described [17]. KRAS codon 12 and 13 mutations, and BRAF codon 600 mutations were detected using PCR-restriction fragment length polymorphism and direct sequencing techniques. Among the 788 CRC samples, 39 and 81 were excluded from the KRAS and BRAF mutation analyses, respectively, due to insufficient DNA quantity.

### 2.6. Mice and Induction of CAC

Induction of CAC in mice was carried out as previously described [18]. Briefly, 5- to 6-week-old C57BL/6 male mice were purchased from OrientBio (Seongnam, Korea) and maintained at the Animal Research Facility at Jeju National University School of Medicine under pathogen-free conditions. Mice (*n* = 32) were intraperitoneally injected with azoxymethane (AOM) (7.4 mg/kg body weight) (Sigma-Aldrich, St Louis, MO, USA) dissolved in phosphate-buffered saline (PBS). After AOM administration, the mice were subjected to 3-week cycles of 1 week with 1% dextran sulfate sodium (DSS treatment) (MP Biomedicals, Santa Ana, CA, USA) added to their drinking water, and 2 weeks of drinking water without DSS (recovery period). The protocol was the same for the control group (*n* = 27) except they received no AOM injections on day 0. Some mice were euthanized prior to DSS treatment and others were euthanized after DSS treatment. Colon tissues were then harvested. Upon opening the colon, the mucosal surface was observed, and tumors were counted. A 1-cm piece of the distal colon was bisected longitudinally; half was stored in RNA*later*^®^ stabilization solution (Ambion, Austin, TX, USA) for real-time PCR analysis and the other half was fixed in a 4% paraformaldehyde neutral buffer solution for histologic examination. Animal experiments were performed in accordance with the institutional guidelines of the Jeju National University for animal use and care.

### 2.7. Tissue Microarray (TMA) Construction

Thirteen TMAs containing 770 CRCs from the SNUH were constructed and histologically evaluated. In brief, the perimeters of representative tumor areas were marked for each sample. Core tissue biopsies (2 mm in diameter) were extracted from each paraffin block (donor blocks) and arranged on a new recipient paraffin block (tissue array block) using a trephine apparatus (SuperBioChips Laboratories, Seoul, Korea). For CRCs from the JNUH, two TMAs containing 24 pairs of adenoma and carcinoma areas and 15 TMAs containing 182 CRCs were generated using 4 mm core tissue biopsies [19]. For ulcerofungating CRCs, both superficial and invading areas were included, and, if present, lymph node metastases were as well. In addition, three TMAs from CAC mouse model animals were constructed as previously described [18]. 

### 2.8. Immunohistochemistry (IHC) and Evaluation

IHC and interpretation was carried out as previously described [15]. IHC was performed on 4 μm TMA sections using a BOND-MAX automated immunostainer and a Bond Polymer Refine Detection kit (Leica Microsystems, Wetzlar, Germany) according to the manufacturer’s instructions. The primary antibodies used were anti-EPHB3 (Novus Biologicals, Littleton, CO, USA; 1B3) and anti-β-catenin (Novocastra Laboratories, Newcastle, UK; 17C2). EPHB3 expression was assessed at the tumor cell membrane. Histo-scores (H-scores; range: 0–300) were obtained by multiplying the intensity score (0 = negative; 1 = weak; 2 = moderate; 3 = strong) and the percentage of positive tumor cells (range: 0–100%). For statistical analyses, TMA sections with H-scores < 40 were defined as negative, and those with H-scores > 40 were defined as positive. β-catenin staining was considered as positive when more than 10% of the tumor cell showed strong nuclear positivity. 

### 2.9. RNA Extraction and Quantitative Real-Time PCR

Total RNA was extracted from 30 paired fresh CRC tissues and matched normal colon tissues with TRIZOL (Invitrogen, Carlsbad, CA, USA). Complementary DNA was generated by reverse transcription using the GoScript reverse transcription system (Promega, Madison, WI, USA). Subsequently, to determine the expression of target genes, real-time PCR was performed with Premix EX Taq (Takara Bio, Shiga, Japan) in a StepOne Plus real-time PCR system (Applied Biosystems, Foster City, CA, USA). The cycling conditions were as follows: initial denaturation for 30 s at 95 °C, followed by 40 cycles of 95 °C for 1 s and 60 °C for 5 s. The TaqMan gene expression assays (Applied Biosystems) used were as follows: Hs00177903-m1 (EPHB3), Hs00362096-m1 (EPHB2), Hs00197437 (OLFM4), Hs00394267_m1 (LRIG1), Hs00173664_m1 (LGR5), Hs01075864_m1 (CD44), Hs01009250-m1 (PROM1/CD133), Hs002379687_s1 (CD24), Hs00233455_m1 (CD166), and Hs0275899_g1 (GAPDH). GAPDH served as the endogenous control.

### 2.10. Colon Cancer Cell Lines

Ten colon cancer cell lines (DLD1, HT29, HCT116, HCT15, SW620, LOVO, SW480, KM12C, KM12L4, and KM12SM) were obtained from the Korean Cell Line Bank (Seoul, Korea). The cells were cultured in MEM, DMEM, RPMI1640, or L15 medium (Welgene, Daegu, Korea) supplemented with 10% fetal bovine serum (FBS) (Gibco, Carlsbad, CA, USA) and 1% penicillin/streptomycin (Gibco), and maintained in a humidified incubator with 5% CO_2_ at 37 °C. 

### 2.11. Antibodies 

The anti-EPHB3 antibody was purchased from Novus Biologicals and anti-β -actin antibody was purchased from Abcam (Cambridge, UK). Anti-EPHB2 was purchased from R&D systems (Minneapolis, MN, USA). The anti-caspase-3, anti-β-catenin, anti-SLUG, anti-AKT, anti-ERK, anti-phospho-AKT, anti-phospho-ERK, anti-cleaved PARP, anti-BAX, and anti-BIM antibodies were purchased from Cell Signaling Technology (Danvers, MA, USA). Anti-mouse IgG-HRP and anti-rabbit IgG-HRP antibodies were purchased from Santa Cruz Biotechnology (Santa Cruz, CA, USA). 

### 2.12. Western Blot Analysis 

Cellular proteins were extracted using lysis buffer (iNtRON Biotechnology, Seongnam, Korea) and proteins were quantitated using BCA protein assay kits (Pierce, Rockford, IL, USA). Cell lysates were run on a 10% SDS-polyacrylamide gel, and proteins were transferred to a PVDF membrane (Millipore Corporation, Bedford, MA, USA) and blocked with 5% non-fat dry milk in PBS-Tween-20 (0.1%, *v/v*) for 1 h. The membranes were then probed with specific primary antibodies. After overnight incubation at 4 °C and washing with TBS containing 0.1% Tween-20, the membranes were incubated for 1 h with corresponding secondary antibodies. The target proteins were visualized with an Alliance-Mini.HD9 chemiluminescence documentation system (UVItec Cambridge, UK).

### 2.13. Transfection of EPHB3 and siRNA

Full-length cDNA encoding EPHB3 (pCMV6-EPHB3) was purchased from Origene (Rockville, MD, USA). Cells (1 × 10^6^ cells/well) were seeded in a 6-well plate and transfected with 5 μg of pCMV-EPHB3, control vector, pCMV-EGFP, or EPHB3 siRNA pool (Dharmacon, Lafyette, CO, USA) using the Invitrogen Neon transfection system. Twenty-four hours post-transfection, the cells were subjected to proliferation and migration assays, RNA was extracted for real-time PCR, and protein was extracted for Western blot analysis. All experiments were carried out independently at least twice.

### 2.14. Proliferation Assay and Colony Formation Assay

For the proliferation assay, 5 × 10^3^ cells/well counted by LUNA-II (Logos Biosystems, Gyeonggi-do, Korea) were seeded in the wells of a 96-well plate and cultured at 37 °C. After adding 10 μL of Cell Counting Kit-8 reagent (Dojindo, Kumamoto, Japan) to each well and incubating for 1 h, optical density was measured at 450 nm in an automatic microplate reader (Thermo Labsystems, Rockford, IL, USA). For the colony formation assay, 5 × 10^3^ cells were seeded in a 60 mm culture dish and cultured for 2–3 weeks until distinguishable colonies appeared. Colonies were fixed with 70% methanol and stained with 0.01–0.1% crystal violet. All experiments were performed in triplicate and repeated at least twice independently.

### 2.15. Migration Assay

For the migration assay we used 24-well transwell culture plates with inserts (8 μm pore size) (BD Bioscience, San Diego, CA, USA). We seeded 2 × 10^5^ cells in 300 μL serum-free medium into the upper chambers, and 500 μL medium with 10% FBS into the lower chambers. After 24 h, cells that remained on the top surface of the transwell insert were carefully wiped away with a cotton swab. The cells that migrated through the pores to the lower surface of the insert were fixed in methanol for 10 min, stained with crystal violet, and counted 1 h later. All experiments were carried out independently at least 2–3 times.

### 2.16. Statistical Analysis

Statistical analyses were performed using SPPSS statistical software version 18.0 (SPSS, Chicago, IL, USA) and Prism version 5.0 (GraphPad Software, San Diego, CA, USA). The association between EPHB3 positivity and clinico-pathological characteristics were assessed using Fisher’s exact test or Pearson’s chi-square test. Between-group comparisons of the real-time PCR data were performed using Student’s *t*-test. The correlations between EPHB3 and candidate cancer stem cell markers were evaluated using the Spearman correlation test. Survival analysis was performed using the Kaplan–Meier and log-rank method. To identify independent prognostic factors, we performed multivariate analyses using the Cox proportional hazards model. A *p*-value < 0.05 was considered statistically significant. 

## 3. Results

### 3.1. EPHB3 Expression in Human Colorectal Cancers and its Correlation with Stem Cell-Related Markers 

To examine the expression of EPHB3, we performed real-time PCR with RNA isolated from fresh-frozen CRC samples and matched non-cancerous colon tissues from 30 patients. The expression level of *EPHB3* mRNA was higher in CRC samples than in non-cancerous tissues in all 30 sample pairs (Figure 1A). The mean *EPHB3* expression was significantly elevated in CRC tissues compared with normal colon tissue (*p* < 0.01) (Figure 1B). EPHB3 protein levels were also higher in cancer tissues than in matched normal tissues in all five cases examined (Figure 1C). As *EPHB3* is one of the genes enriched in intestinal stem cells, we measured the expression of other intestinal stem cell markers including *EPHB2*, *OLFM4*, *LRIG1*, and *LGR5*, and examined if their expression patterns correlated with *EPHB3*. Notably, *EPHB3* expression was positively associated with the expression of *EPHB2* (*p* < 0.0001), *OLFM4* (*p* = 0.02), and *LRIG1* (*p* = 0.002) (Figure 1D). Several other molecules, including *CD44*, *CD133*, *CD24*, and *CD166* are potential cancer stem cell markers that may contribute to CRC progression and metastasis [20]; therefore, we assessed if their expression correlated with that of *EPHB3* in the CSC samples and controls. We found that *CD44* was significantly associated with *EPHB3* (*p* = 0.02) (Figure 1E). 

### 3.2. EPHB3 Expression in Colorectal Precancerous Lesions

Immunohistochemical analysis for EPHB3 expression was conducted with normal small (*n* = 1) and large (*n* = 5) intestine, and various precancerous lesions including hyperplastic polyps (HPs, *n* = 5), sessile serrated adenomas (SSAs, *n* = 5), traditional serrated adenomas (TSAs, *n* = 3), and conventional tubular adenomas with low grade dysplasia (TAs, *n* = 5). In normal small (Figure 2A) and large (Figure 2B) intestinal mucosa, EPHB3-expressing cells were confined to the base of the crypts, confirming EPHB3 as an intestinal stem cell marker. Similar to normal intestinal mucosa, EPHB3 expression in HPs and SSAs was limited to the base of intestinal crypts (Figure 2C,D). TSAs showed a distinctive expression pattern; EPHB3 was expressed in the basal areas as well as in the villi, which harbored ectopic crypt foci (ECF). Interestingly, EPHB3 expression was prominent at the bases of the ECFs (Figure 2E). Conventional TAs exhibited strong, diffuse EPHB3 staining (Figure 2F) and showed markedly higher H-scores compared to other lesions (*p* = 0.004) (Figure 2G). 

### 3.3. Expression Profile of EPHB3 During CRC Progression

To further investigate the change in EPHB3 expression during CRC progression, we identified 22 CRC samples that included regions of carcinoma and of pre-existing adenomas. EPHB3 expression was significantly lower in the carcinoma regions (mean H-score: 58) than in the adenoma regions (mean H-score: 110) of the samples (*p* = 0.02) (Figure 3A,B). However, EPHB3 expression completely disappeared during the adenoma–carcinoma transition in only one case (Figure 3A). Subsequently, we identified 37 cases of ulcerofungating CRC that had metastasized to the regional lymph nodes. To determine the changes in EPHB3 expression during invasion and lymph node metastasis, we compared H-scores in three regions of each CRC case: a superficial fungating region, the invasive fronts, and metastatic cancer cells in the lymph nodes. Interestingly, H-scores were significantly lower in the invasive fronts than in fungating areas, but no significant difference was observed between EPHB3 expression in the invasive fronts and the lymph node metastases (Figure 3C,D). These results indicate that EPHB3 downregulation occurs mainly at two points: the transformation from adenoma to carcinoma, and invasion into deeper tissue layers.

### 3.4. Decline of EPHB3 Expression in Budding Cancer Cells at the Invasive Front 

Cancer budding, a risk factor for metastasis and worse prognosis, is frequently detected at the invasive front of CRCs [21], thus we sought to determine whether EPHB3 expression is altered in the budding cells. Remarkably, H-scores of EPHB3 in the budding cells were significantly lower compared with non-budding cells (*n* = 11, *p* < 0.001) (Figure 4A,B). As E-cadherin dysregulation is frequently observed in the budding cells of the invasive front, we compared EPHB3 and E-cadherin expression in the serial sections. As expected, in budding cancer cells both EPHB3 and E-cadherin expression were downregulated. Notably, EPHB3 suppression at the invasive front was more extensive than E-cadherin (Figure 4C), suggesting that EPHB3 expression is inhibited earlier than E-cadherin. Next, to examine whether EPHB3 might modulate E-cadherin levels, we induced EPHB3 in DLD1 cells, which have negligible levels of endogenous EPHB3 and E-cadherin, and found that exogenous overexpression of EPHB3 also enhanced E-cadherin expression (Figure 4D). Furthermore, reducing E-cadherin in HT29 cells using siRNAs slightly decreased EPHB3 expression and increased phospho-ERK levels, while levels of EPHB2, the closest homologue of EPHB3, were not altered (Figure 4E). Based on these findings, we hypothesize that EPHB3 suppression occurs earlier than E-cadherin dysregulation, and may in part, contribute to the downregulation of E-cadherin.

### 3.5. EPHB3 Expression in CACs

CACs have distinct histologic and molecular characteristics different from conventional CRCs that develop through adenoma–carcinoma progression [22,23]. Therefore, we explored the *EPHB3* expression profile in a mouse model of CAC induced by AOM and DSS treatment. Distal colon was harvested before and after each DSS treatment, and consistent with our previous study, mice treated with AOM and DSS only developed visible colon tumors after three cycles of DSS treatment (Figure 5A) [18]. Real-time PCR analysis revealed increased *EPHB3* expression at days 65 and 71 in the AOM/DSS-treated mice (Figure 5B). RNA in situ hybridization for *EPHB3* with tissue arrays containing colon samples of normal (untreated), DSS-treated, and AOM/DSS-treated mice demonstrated higher *EPHB3* expression in regenerative glands, microtumors, and macrotumors compared to normal colon mucosa (Figure 5C,D). Although we observed enhanced *EPHB3* expression in the colitis-associated tumors, we did not detect the *EPHB3* downregulation typical of CRC progression in human samples probably because overt adenoma to carcinoma transition does not occur in this murine model of CAC, and invasive carcinoma is extremely rare. 

### 3.6. Clinico-Pathological and Prognostic Significance of EPHB3 Expression in CRCs

IHC for EPHB3 was performed on 13 tissue microarrays. Using an H-score of > 40 to define EPHB3 positivity, 147 CRC tissue samples (24%) were positive for EPHB3. Representative EPHB3-negative and -positive cases are shown in Figure 6A. The association between EPHB3 expression and clinico-pathological factors are summarized in Table 1. EPHB3 positivity was significantly higher in well-differentiated (*p* = 0.002) and lower stage (*p* < 0.001) CRCs. Mucin production (*p* = 0.042) and elevated levels of tumor infiltrating lymphocytes ((TIL) > 8 per high power field) (*p* < 0.001) were more frequently observed in EPHB3-positive CRCs. Lymphatic invasion (*p* < 0.001) and venous invasion (*p* = 0.004) were negatively associated with EPHB3 expression. No significant association was seen between EPHB3 and nuclear β-catenin expression, a marker of Wnt pathway activation. Interestingly, EPHB3 positivity was more frequently observed in MSI-high CRCs but was not associated with CIMP status or mutations in KRAS or BRAF (Table 2). Kaplan–Meyer analysis demonstrated that CRC patients with EPHB3 positivity have better clinical outcomes in both overall (*p* = 0.007) and recurrence-free survival (*p* < 0.001) (Figure 6B) and the prognostic impact of EPHB3 was apparent in stage III CRC patients (Figure 6C); however, in a multivariate analysis EPHB3 was not an independent prognostic marker (HR = 0.832, *p* = 0.400) (Table 3).

### 3.7. Effects of EPHB3 Expression on the Growth and Migration of CRC Cells

We screened 12 CRC cell lines and found that DLD1 cells expressed extremely low levels of the *EPHB3* transcript and EPHB3 protein (Figure 7A). To investigate the functional roles of EPHB3 in the regulation of CRC growth and migration, we overexpressed EPHB3 in DLD1 cells by transfecting them with an EPHB3 expression plasmid DNA. The growth rate of DLD1 cells transfected with EPHB3 was attenuated compared to those transfected with a control plasmid (Figure 7B). As phosphoinositide-3-kinase–protein kinase B/AKT (PI3K-PKB/AKT) and mitogen-activated kinase (MAPK) signaling pathways are often involved in cancer cell proliferation [24], we examined the impact of EPHB3 expression on the activation of the AKT and ERK proteins. Upon EPHB3 expression, our immunoblot results showed that phospho-ERK levels decreased significantly in DLD1 cells, whereas phospho-AKT levels did not change (Figure 7C). In addition, cleaved caspase-3 and PARP levels increased without significant alterations in BAX and BIM expression (Figure 7C). These findings suggest that EPHB3-induced growth inhibition in DLD1 cells may be due to increased apoptosis resulting from downregulation of the MAPK pathway. Furthermore, EPHB3 overexpression resulted in a reduced colony-forming ability and reduced migration of DLD1 cells (Figure 7D,E). Taken together, these results demonstrate that EPHB3 suppresses the proliferation and migration of CRC cells.

## 4. Discussion

In this study, we investigated the expression profile of EPHB3 in various precancerous lesions and in human CRCs. Although EPHB3 expression in CRCs has been reported by previous studies, to our knowledge, this is the most comprehensive study to demonstrate the alterations of EPHB3 expression throughout CRC development and progression. Using a large cohort of CRC patients for whom we had tumoral histopathological and molecular data, we found that EPHB3 expression is associated with improved clinical outcomes. This is consistent with its role as a tumor suppressor, controlling cellular positioning and restricting tumor cell motility [9,25].

In accordance with its role as a stem cell-related gene in the intestinal crypts, the expression of *EPHB3* mRNA positively correlated with multiple intestinal stem cell (ISC) markers including *EPHB2, OLFM4,* and *LRIG1* (Figure 1D), but not with *LGR5*. In addition, EPHB3 protein expression closely correlated with that of its homologue, EPHB2 (Figure 6B). These results may imply that close associations among ISC signature genes represent a stem cell hierarchy that is maintained during colorectal carcinogenesis. As *CD44* only showed a positive correlation with *EPHB3* among the candidate CSC markers we examined, it would be interesting to explore the interplay between CD44 and EPHB3 and its effect on cancer stem cell biology in CRCs in future studies. 

Different molecular pathways contribute to the development of CRCs including chromosomal instability (CIN), MSI, and CIMP. MSI-high CRCs usually occur in the proximal colon. They have a high *BRAF* mutation rate, and they are characterized by lymphocytic infiltration, mucin secretion, and poor differentiation [26]. Interestingly, EPHB3 positivity was significantly higher in MSI-high CRCs than in MSI-low or negative ones (Table 2). We also found that mucin production and lymphocytic infiltration was significantly higher in EPHB3-positive CRCs, and although it did not reach statistical significance, EPHB3-positive CRCs appeared more frequently in the proximal colon than in the distal (Table 1). These findings suggest that the MSI-high phenotype may have an impact on EPHB3 expression. Inconsistent with this hypothesis, we found no association between EPHB3 and BRAF mutations, and lower EPHB3 positivity in poorly differentiated CRCs, which makes it more complicated to characterize the link between MSI and EPHB3 in CRCs.

In the normal small and large intestines, EPHB3-positive cells were restricted to the crypt bases, supporting the notion that EPHB3 may be an ISC marker. In HPs and SSAs, EPHB3 was expressed at the base of the crypts, whereas in TSAs and TAs, EPHB3 expression was no longer confined to the crypt bases. Notably, in TSAs, EPHB3 expression was frequently observed in ECF as spots in which a group of cells with stem cell-like phenotypes exist and expand. Compared with other lesions, TAs exhibited a much stronger, diffuse EPHB3 expression. This may be because most conventional TAs develop through upregulated Wnt signaling due to an *APC* gene mutation, and EPHB3 is a direct target of Wnt signaling [7]. However, when comparing the expression of EPHB3 and nuclear β-catenin, which reflects aberrant Wnt activation, in CRCs, no significant association was observed. The independent functioning of the EPHB3 and Wnt signaling pathways in CRCs may be due to epigenetic regulation or the involvement of other pathways. The transcriptional enhancement of EPHB3 is influenced by the intestinal stem-cell regulator, ASCL2, MAP kinase, and Notch signaling, in addition to Wnt/ β-catenin signaling. In fact, elevated Notch activity in CRCs leads to enhancer dysfunction and EPHB3 silencing [26]. Additionally, in gastric cancers EZH2, a methyltransferase and the core catalytic subunit of polycomb repressive complex 2, may mediate the epigenetic suppression of EPHB3 [27,28]. 

We found that enhanced EPHB3 expression in adenoma significantly decreased during the transformation to carcinoma, and it declined further when cancer cells invaded into the muscular layers. These findings indicate that EPHB3 expression increases during the early stages of tumorigenesis but is suppressed as tumors progress. This biphasic expression pattern has been reported for other intestinal stem cell markers such as LGR5 and EPHB2 [15,19]. In addition, EPHB3 expression was substantially reduced in the budding cells at the invasive front of tumors. As E-cadherin is frequently downregulated in budding cells and is a crucial step that promotes tumor invasion and metastasis [29], we investigated whether EPHB3 influences E-cadherin expression and vice versa. Notably, E-cadherin suppression by siRNAs in DLD1 cells led to the downregulation of EPHB3, but it had no effect on EPHB2 expression. Additionally, EPHB3 suppression led to the decline of E-cadherin expression. Since E-cadherin and EPHB3 are tumor suppressors, the concomitant reduction of these two proteins in budding cells at the invasive tumor front may significantly contribute to the migration of these cancer cells. Regarding the association of EPHB3 with the epithelial-to-mesenchymal transition (EMT), Ronsch et al. demonstrated that SNAIL1, an EMT regulator, silences EPHB3 by disabling a transcriptional enhancer element to facilitate the EMT in several CRC cell lines [30]. However, whether EMT regulators are typically involved in the budding of CRCs is controversial because recent studies using immunohistochemistry and RNA in situ hybridization on human CRC samples neglected to find the expression of any EMT transcription factors by cancer cells at the invasive front of CRC tumors [19,31]. Further studies are required to identify the molecular mechanisms (including signaling pathways) that govern EPHB3 silencing at the invasive tumor fronts. 

To investigate the underlying mechanism of EPHB3 as an effective prognostic marker in CRC patients, we transfected DLD1 cells with an EPHB3 expression plasmid and found that overexpression of EPHB3 attenuated the growth and migration of these cells. In addition, upon EPHB3 transfection, phospho-ERK levels decreased significantly but phospho-AKT levels were not altered. In addition, cleaved caspase 3 and PARP levels increased, indicating that the MAPK pathway may be involved in the EPHB3-induced growth suppression by enhancing apoptosis, which suggests a pro-apoptotic role for EPHB3. On the other hand, in non-small cell lung cancers, EPHB3 overexpression leads to decreased AKT activity, which suppresses tumor cell migration and metastasis [13]. These data indicate that the mechanism of EPHB3-mediated tumor suppression may differ by cancer type. Additionally, since BAX levels remained the same and BIM expression decreased only slightly upon EPHB3 transfection, it is unlikely that EPHB3-induced programmed cell death is mediated by the intrinsic apoptotic pathway and the Bcl-2 family of proteins.

## 5. Conclusions 

Using tumor samples from a large cohort of CRC patients we demonstrated that EPHB3 expression is positively associated with less advanced tumor stages and therefore better clinical outcomes. Enhanced EPHB3 expression in TAs declined during the transformation from adenoma to carcinoma, and it further decreased when the tumor began invading deeper into the muscular layers. Loss of EPHB3 expression was frequently observed in budding cancer cells at the invasive front of CRC tumors, and this loss appears to precede E-cadherin downregulation. Moreover, in vitro studies showed that EPHB3 overexpression attenuated the proliferation and migration of colon cancer cells, and this was probably due to reduced MAP kinase signaling. These findings suggest that EPHB3 may act as a tumor suppressor in late stages of CRC progression.

## Figures and Tables

**Figure 1 biomolecules-10-00602-f001:**
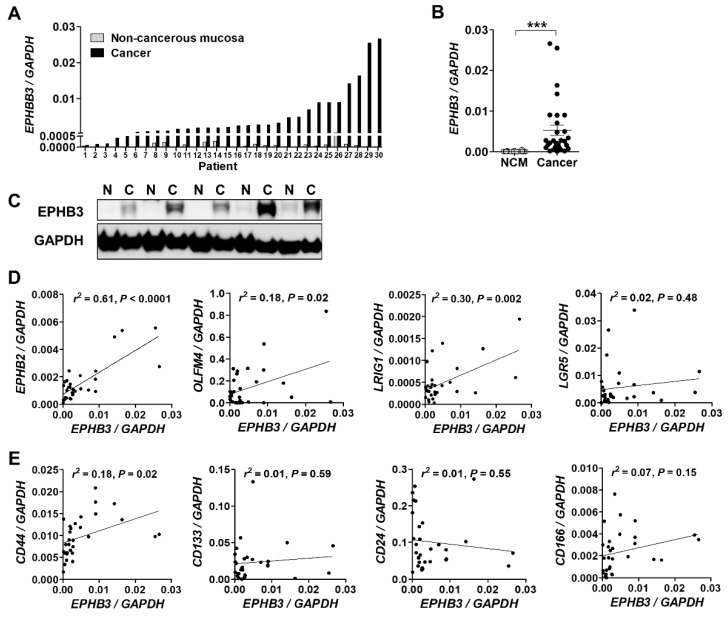
The expression of intestinal and cancer stem cell markers, and Ephrin type-B receptor 3 (EPHB3) in human colorectal cancer samples. Real-time PCR analysis of the expression of intestinal stem cell markers (*EPHB2, OLFM4, LRIG1,* and *LGR5*) and cancer stem cell markers (*CD44, CD133, CD24,* and *CD166*) from 32 pairs of fresh-frozen colorectal cancers (CRCs) samples and matched, non-cancerous mucosa (NCM). (**A**,**B**) *EPHB3* expression level was significantly higher in cancers than in NCM. Data are presented as the mean ± SEM. SEM: standard error of the mean. (**C**) Expression of EPHB3 in normal and colon cancer tissues by immunoblotting. (*n* = 6) (**D**) EPHB3 expression positively correlated with EPHB2, OLFM4, and LRIG1, but not with LGR5. (**E**) *EPHB3* expression also positively correlated with *CD44* expression. N, normal; C, cancer. ^***^
*p* < 0.001.

**Figure 2 biomolecules-10-00602-f002:**
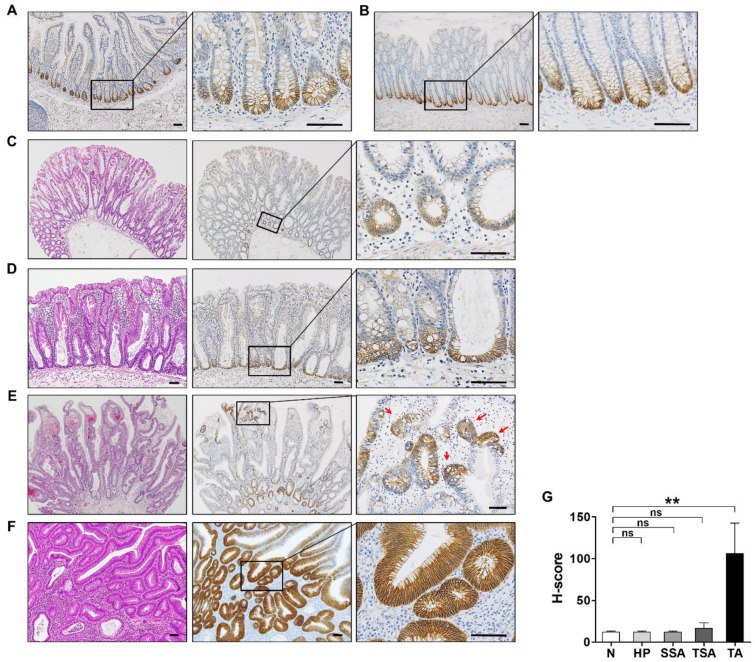
Distinct EPHB3 expression patterns in colorectal precancerous lesions. Representative images of EPHB3 expression in normal small (**A**) (*n* = 1) and large (**B**) (*n* = 5) intestinal mucosa, hyperplastic polyp (HP, *n* = 5) (**C**), sessile serrated adenoma (SSA, *n* = 5) (**D**), traditional serrated adenoma (TSA, *n* = 3) (**E**), and conventional tubular adenoma (TA, *n* = 5) (**F**). (**G**) Histo-scores of EPHB3 in each lesion. Data are presented as the mean ± SEM. SEM: standard error of the mean. ns, not significant. ** *p* < 0.01. Scale bars: 50 μm.

**Figure 3 biomolecules-10-00602-f003:**
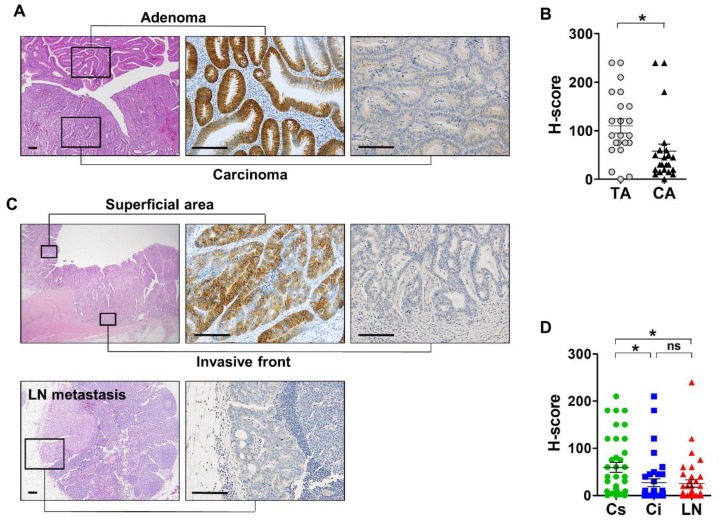
Decline of EPHB3 expression during colorectal cancer progression. (**A**,**B**) In colorectal cancers arising within an adenoma background (*n* = 22), EPHB3 expression was significantly lower in carcinoma-rich regions than in adenoma-rich regions. (**C**,**D**) EPHB3 expression was examined in ulcerofungating cancers with lymph node metastases (*n* = 37). Histo-scores of EPHB3 gradually decreased when cancer cells invaded into muscle layers and metastasized to lymph nodes. Data are presented as the mean ± SEM. SEM: standard error of the mean. ns, not significant. * *p* < 0.05. Scale bars: 50 μm.

**Figure 4 biomolecules-10-00602-f004:**
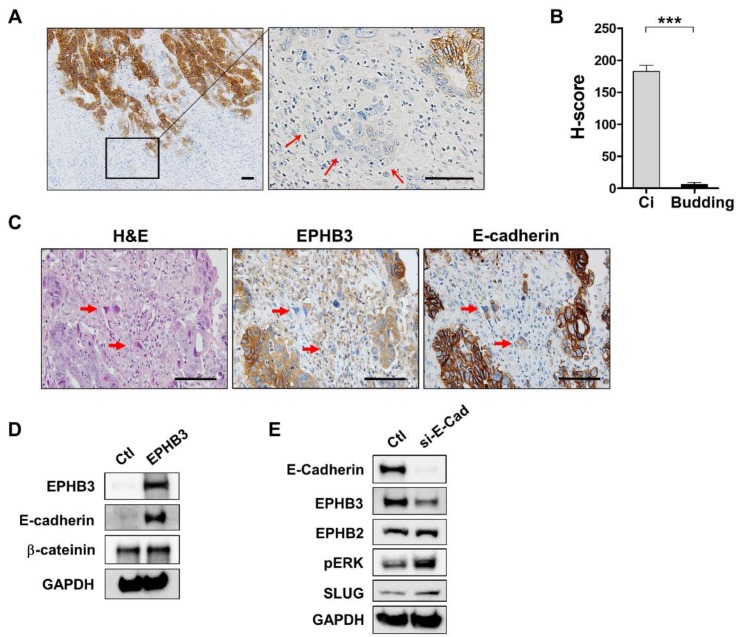
EPHB3 is downregulated in budding cancer cells at the invasive front of CRC tumors. (**A**,**B**) EPHB3 expression markedly declined in the budding cells at the invasive front of colon cancer tumors (*n* = 11), (red arrows). (**C**) At the invasive tumor front, EPHB3 expression decreased more significantly than E-cadherin. Red arrows indicate budding tumor cells that are negative for EPHB3 and E-cadherin. (**D**) EPHB3 overexpression induced E-cadherin expression in DLD1 cells. (**E**) In HT29 cells, E-cadherin suppression by siRNA resulted in the downregulation of EPHB3. Ctl, control; Ci, cancer at invasive fronts; E-cad, E-cadherin. *** *p* < 0.001. Scale bars: 50 μm.

**Figure 5 biomolecules-10-00602-f005:**
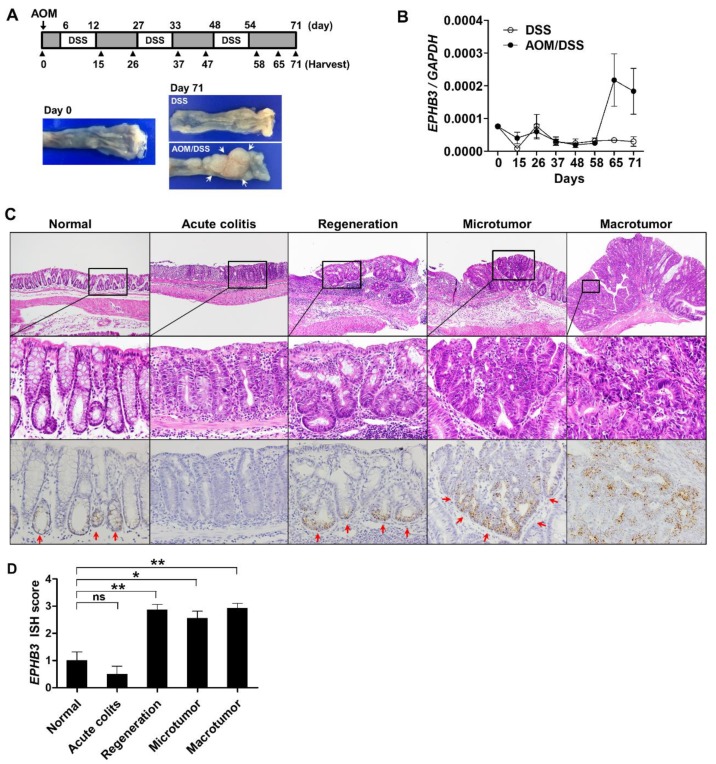
Expression of EPHB3 in colitis-associated carcinogenesis (CAC). (**A**) Colitis-associated colon cancer was induced by azoxymethane and dextran sodium sulfate (AOM/DSS). At each day indicated by black arrow heads, mice (DSS: *n* = 4, AOM/DSS: *n* = 5) were sacrificed for analysis. Representative photos of colons harvested on day 0 and day 71 show visible tumors indicated by white arrows. (**B**) Real-time PCR analysis revealed increased EPHB3 expression on days 65 and 71. (**C**,**D**) RNA in situ hybridization for EPHB3 expression during colitis-associated colon cancer development. Data represent the means ± SEM. SEM: standard error of the mean. * *p* < 0.05, ** *p* < 0.01.

**Figure 6 biomolecules-10-00602-f006:**
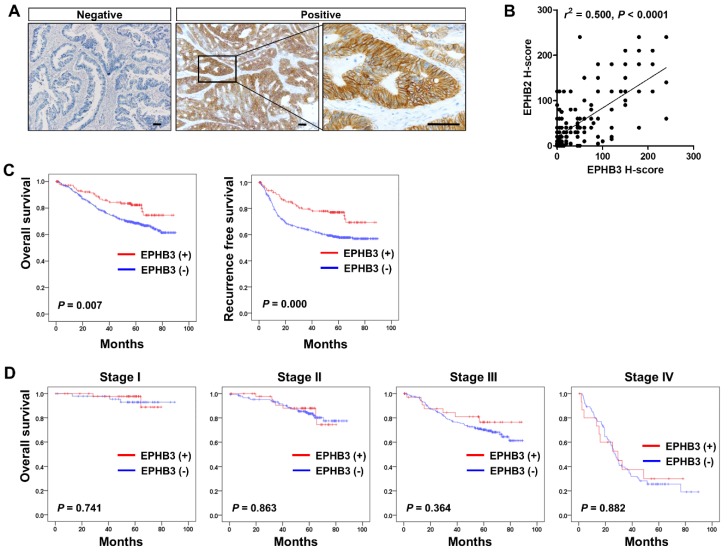
There is a positive correlation between EPHB3 expression and better clinical outcomes of patients with colorectal cancer (CRC). Immunohistochemistry for EPHB3 was performed in a large cohort of CRCs (*n* = 610). (**A**) Representative cases of CRCs that are negative or positive for EPHB3. (**B**) There was a strong correlation in the H-scores of EPHB2 and EPHB3. (**C**) Kaplan–Meier analysis demonstrated that EPHB3 positivity showed better overall (*p* = 0.007) and recurrence-free (*p* = 0.000) survival rates. (**D**) Prognostic significance of EPHB3 expression was more apparent in stage III CRC patients. Scale bars: 50 μm.

**Figure 7 biomolecules-10-00602-f007:**
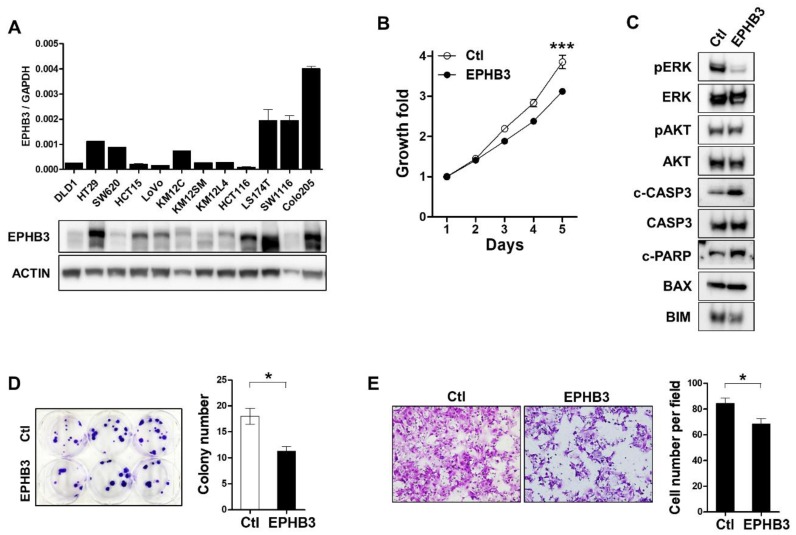
Suppressive effects of EPHB3 on colorectal cancer (CRC) cell growth and migration. (**A**) The mRNA and protein levels of EPHB3 in CRC cell lines were assessed by real-time PCR and immunoblot assays. (**B**) DLD1 was transfected with a control or an EPHB3 expression plasmid, and cells were counted to assess proliferation. (**C**) Twenty-four hours after transfection, immunoblot assays were performed using the indicated antibodies. (**D**) Colony formation assays were performed by counting the number of colonies from control or EPHB3-overexpressing DLD1 cells. (**E**) The effect of EPHB3 expression on the migration of DLD1 cells by transwell migration assay. * *p* < 0.05, *** *p* < 0.001. Ctl, control.

**Table 1 biomolecules-10-00602-t001:** Association between EPHB3 expression and the clinicopathological characteristics.

Characteristics	Total (%)	EPHB3		*p*-Value
		Negative (%)	Positive (%)	
Patients	610 (100)	463 (76)	147 (24)	
Age				
≥60	345 (57)	254 (74)	91 (26)	0.152 ^†^
<60	265 (43)	209 (79)	56 (21)	
Gender				
Female	240 (39)	178 (74)	62 (26)	0.439 ^†^
Male	370 (61)	285 (77)	85 (23)	
Location				
Proximal	161 (26)	115 (71)	46 (29)	0.133 ^†^
Distal	449 (74)	348 (78)	101 (22)	
Differentiation				
WD	46 (8)	26 (57)	20 (44)	0.002 ^#^
MD	544 (89)	419 (77)	125 (23)	
PD	20 (3)	18 (90)	2 (10)	
Mucin				
Absent	539 (88)	416 (77)	123 (28)	0.042 ^#^
Present	71 (12)	47 (66)	24 (34)	
TIL > 8				
Negative	462 (76)	369 (80)	93 (20)	<0.001 ^†^
Positive	148 (24)	94 (64)	54 (37)	
Lymphatic invasion				
Negative	341 (56)	232 (68)	109 (32)	<0.001 ^†^
Positive	269 (44)	231 (86)	38 (14)	
Venous invasion				
Negative	526 (86)	389 (74)	137 (26)	0.004 ^†^
Positive	84 (14)	74 (88)	10 (12)	
TNM_7th				
I	95 (156)	46 (48)	49 (52)	<0.001 ^#^
II	197 (32)	149 (76)	48 (24)	
III	221 (36)	187 (85)	34 (15)	
IV	97 (16)	81 (84)	16 (16)	
β-catenin				
No nuclear stain	225 (37)	172 (76)	53 (24)	0.845 ^†^
Nuclear stain	385 (63)	291 (76)	94 (24)	

^†^ Fisher’s exact test; ^#^ Pearson chi-square test; TIL, tumor infiltrating lymphocytes.

**Table 2 biomolecules-10-00602-t002:** Association between EPHB3 expression and molecular characteristics.

Characteristics	Total (%)	EPHB3		*p*-Value
		Negative (%)	Positive (%)	
Patients	610 (100)	463 (76)	147 (24)	
CIMP				
Negative and Low	570 (93)	429 (75)	141 (25)	0.185 ^†^
High	40 (7)	34 (85)	6 (15)	
MSI				
Negative and Low	559 (92)	434 (78)	125 (22)	0.002 ^†^
High	51 (8)	29 (57)	22 (43)	
	Total (%)	EPHB3		*p*-Value
		Negative (%)	Positive (%)	
Patients	590 (100)	447 (76)	143 (24)	
KRAS				
Wt	432 (73)	329 (76)	103 (24)	0.745 ^†^
Mt	158 (27)	118 (75)	40 (25)	
	Total (%)	EPHB3		*p*-Value
		Negative (%)	Positive (%)	
Patients	605 (100)	458 (76)	147 (24)	
BRAF				
Wt	572 (95)	430 (75)	142 (25)	0.296 ^†^
Mt	33 (5)	28 (85)	5 (15)	

^†^ Fisher’s exact test; ^#^ Pearson chi-square test; CIMP, CpG island methylator phenotype; MSI, microsatellite instability; Wt, wild type; Mt, mutation.

**Table 3 biomolecules-10-00602-t003:** The results of univariate and multivariate analysis for survival rate in colorectal cancers.

Variables	Category	Univariate Analysis			Multivariate Analysis		
		HR	95% CI	*p*-Value	HR	95% CI	*p*-Value ^a^
Age	>60/<60	1.271	1.008–1.602	0.043	1.668	1.215–2.288	0.002
Gender	Female/male	0.974	0.774–1.227	0.825			
Site	Distal/proximal	0.7	0.549–0.894	0.004	0.742	0.526–1.047	0.090
Differentiation	Poor/moderate/well	3.31	2.280–4.805	0.000	1.366	0.807–2.311	0.245
Lymphatic invasion	Positive/negative	2.917	2.309–3.685	0.000	1.122	0.810–1.554	0.489
Venous invasion	Positive/negative	3.202	2.487–4.123	0.000	1.845	1.306–2.607	0.001
TIL > 8	Positive/negative	0.682	0.465–1.001	0.050	0.735	0.495–1.090	0.126
Mucin	Present/absent	1.048	0.670–1.640	0.836			
Stage	IV/III/II/I	3.476	2.974–4.064	0.000	2.758	2.228–3.414	<0.001
KRAS mutation	Present/absent	1.05	0.745–1.480	0.779			
BRAF mutation	Present/absent	1.307	0.709–2.409	0.390			
CIMP	High/low or negative	1.964	1.205–3.201	0.007	0.735	0.495–1.090	0.423
MSI	Unstable/stable	0.865	0.481–1.554	0.627			
Nuclear β-catenin	Positive/negative						
EPHB3	Positive/negative	0.568	0.374–0.862	0.008	0.832	0.541–1.278	0.400

HR, hazard ratio; CI, confidence interval; TIL, tumor infiltrating lymphocytes; CIMP, CpG island methylator phenotype; MSI, microsatellite instability. ^a^ Cox proportional hazard model.

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
