# Peer review of "Expression Profile and Prognostic Significance of EPHB3 in Colorectal Cancer"

_biomolecules, 2020, doi:10.3390/biom10040602_

Round 1

Reviewer 1 Report

In this manuscript Bo Gun Jang, et al. described results of their work aimed at evaluation of some essential features of Ephrin type-B receptor 3 (EPHB3) expression that are important in development of colorectal cancer (CRC). The authors demonstrated that in CRC like in other tumors EPHB3 play a tumor suppressive role. In particular, the results of experiments with colon cancer cell lines showed that EPHB3 overexpression decreased significantly proliferation and migration of the cells. The analysis of human CRC tissue samples revealed considerable decline of EPHB3 expression in the process of adenoma to carcinoma transformation. Additionally, the obtained data demonstrated loss of EPHB3 expression in many budding tumor cells at invasive front of CRC. Taken together, the study indicates that EPHB3 is a biomarker and prognostic factor for patients with colorectal carcinoma; increased EPHB3 expression is associated with better clinical outcomes.

Manuscript is well written and the conclusions are convincingly supported by experimental results. I have no suggestion to improve the quality of the manuscript.

Author Response

We thank the reviewer for careful evaluation of our manuscript. 

Reviewer 2 Report

Congratulations for this excellent and comprehensive study.  This manuscript is very well prepared ad valuable.

Just three minor comments:

In the conclusions there is not fully justified sentence “Using tumor samples from a large cohort of CRC patients we demonstrated that EPHB3 expression is positively associated with better clinical outcomes”. I propose the following sentence  or similar:” Using tumor samples from a large cohort of CRC patients we demonstrated that EPHB3  expression is positively associated with less advanced tumor stages and therefore  better clinical outcomes. “ This will better respond the results.
There are some  not typical symbols instead of  Greek letters β and µ, but not in every case.
It would be good to define proximal and distal  location.

As a kind of suggestion, but not the required change, it would be interesting to look what is the influence of EPHB3 expression on survival in left sided and right sided CRC separately.

Author Response

We really appreciate your careful review and we tried to address the issues you raised as follows.

1. In the conclusions there is not fully justified sentence “Using tumor samples from a large cohort of CRC patients we demonstrated that EPHB3 expression is positively associated with better clinical outcomes”. I propose the following sentence or similar:” Using tumor samples from a large cohort of CRC patients we demonstrated that EPHB3 expression is positively associated with less advanced tumor stages and therefore  better clinical outcomes. “This will better respond the results. 

: As suggested, we changed the sentence to “Using tumor samples from a large cohort of CRC patients we demonstrated that EPHB3 expression is positively associated with less advanced tumor stages and therefore better clinical outcomes.” (Line 14, p19)

2. There are some not typical symbols instead of Greek letters β and µ, but not in every case.

: We fixed the errors about “β” in 4 spots. (line 4, p5; line 4, p13; line 21, p18; line 26, p18)

3. It would be good to define proximal and distal location.

: We included the definition of proximal and distal colon in the section of material and method. (Line 35-37, p2)

“With regard to tumor location, proximal colon was defined as proximal to the splenic flexure (cecum, ascending and transverse colon) and distal colon was defined as distal to the splenic flexure (splenic flexure, descending, sigmoid colon and rectum.”

4. As a kind of suggestion, but not the required change, it would be interesting to look what is the influence of EPHB3 expression on survival in left sided and right sided CRC separately.

: Thanks for your suggestion, and we did check out on the impact of EPHB3 on survival in left-sided and right-sided CRC and found that the prognostic significance of EPHB3 was more apparent in right-sided CRC (P = 0.029) than in left-sided (0.053).